# Exploring the Impact of Herbal Therapies on COVID-19 and Influenza: Investigating Novel Delivery Mechanisms for Emerging Interventions

Lucas Fornari Laurindo [1,2], Ledyane Taynara Marton [2], Giulia Minniti [2], Victória Dogani Rodrigues [1], Rodrigo Buzinaro Suzuki [2,3], Virgínia Maria Cavallari Strozze Catharin [2,4], Rakesh Kumar Joshi [5] and Sandra Maria Barbalho [2,4,6,*]

1 Department of Biochemistry and Pharmacology, School of Medicine, Faculdade de Medicina de Marília (FAMEMA), Avenida Monte Carmelo, 800, Marília 17519-030, SP, Brazil; lucasffffor@gmail.com (L.F.L.)
2 Department of Biochemistry and Pharmacology, School of Medicine, University of Marília (UNIMAR), Avenida Hygino Muzzy Filho, 1001, Marília 17525-902, SP, Brazil
3 Department of Microbiology and Parasitology, School of Medicine, Faculdade de Medicina de Marília (FAMEMA), Avenida Monte Carmelo, 800, Marília 17519-030, SP, Brazil
4 Postgraduate Program in Structural and Functional Interactions in Rehabilitation, University of Marília (UNIMAR), Avenida Hygino Muzzy Filho, 1001, Marília 17525-902, SP, Brazil
5 Department of Education, Government of Uttarakhand, Nainital 263001, India
6 Department of Biochemistry and Nutrition, School of Food and Technology of Marília (FATEC), Avenida Castro Alves, 62, Marília 17500-000, SP, Brazil
* Correspondence: smbarbalho@gmail.com

**Abstract:** Synthetic antivirals and corticosteroids have been used to treat both influenza and the SARS-CoV-2 disease named COVID-19. However, these medications are not always effective, produce several adverse effects, and are associated with high costs. Medicinal plants and their constituents act on several different targets and signaling pathways involved in the pathophysiology of influenza and COVID-19. This study aimed to perform a review to evaluate the effects of medicinal plants on influenza and COVID-19, and to investigate the potential delivery systems for new antiviral therapies. EMBASE, PubMed, GOOGLE SCHOLAR, and COCHRANE databases were searched. The studies included in this review showed that medicinal plants, in different formulations, can help to decrease viral spread and the time until full recovery. Plants reduced the incidence of acute respiratory syndromes and the symptom scores of the illnesses. Moreover, plants are related to few adverse effects and have low costs. In addition to their significance as natural antiviral agents, medicinal plants and their bioactive compounds may exhibit low bioavailability. This highlights the need for alternative delivery systems, such as metal nanoparticles, which can effectively transport these compounds to infected tissues.

**Keywords:** medicinal plants; antiviral; SARS-CoV-2; COVID-19; influenza; delivery systems; nanomedicine; nanocarriers; antiviral therapies

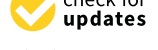



## 1. Introduction

SARS-CoV-2 is a new coronavirus line identified in China at the end of 2019. This virus's origin and fast spread were referred to as a spillover process associated with the consumption of wild animals. This viral agent presents high transmissibility, even in asymptomatic and convalescent periods, leading to a quick spread worldwide and reaching devastating pandemic proportions. This condition has multiple manifestations, from non-symptomatic to mild, moderate, or severe, leading to a multi-organ failure and the inevitable death of millions of people worldwide [1–5].

Another viral condition is influenza, which, similarly to COVID-19, affects the respiratory tract due to direct viral infection or due to an imbalanced response in the immune system. These viruses belong to the orthomyxovirus family. The envelope of the influenza A virus exhibits two surface glycoproteins named hemagglutinin A (responsible for membrane fusion) and neuraminidase (responsible for the release of the virions) [6–8]. H1N1 and H3N2 are subtypes of influenza A that seem to mutate during the season and are associated with a more severe condition, when compared with influenza B [9–12]. Viral infections related to H1N1 have been reported worldwide and are a significant cause of severe acute pneumonia and acute respiratory distress syndrome (ARDS). Authors estimate that about five million cases of severe illness and about 290,000 to 650,000 deaths are due to influenza infection every year [8,13–16]. For these reasons, H1N1 epidemics and COVID-19 represent an unquestionable burden to public health systems [17,18].

Multiple therapies have been developed for years for the prevention of or a therapeutic approach to influenza, and many have been investigated in the last two years for the prevention or treatment of COVID-19. These therapies present various possibilities, including antivirals, corticosteroids, anticoagulants, renin-angiotensin system inhibitors, and several others [19–21]. The treatment approach for COVID-19 is challenging due to the rapid appearance of mutant strains, virus adaptation, and resistance to antiviral drugs. Moreover, the high costs and relevant side effects of antiviral drugs should be considered. As for influenza, there is a general interest in developing a safe and effective vaccine against this pathology [22,23].

Furthermore, medicinal plants and their phytocompounds have been investigated [24–27]. Traditional Chinese herbal medicine attracted international attention during COVID-19, combining a mix of herbal plants and different bioactive compounds. This can act on several distinct targets and signaling pathways, leading to relevant results in treating viral respiratory conditions [28–32]. Medicinal plants present a plethora of bioactive compounds that have been used since ancient times to treat many diseases, such as flavonoids, alkaloids, anthraquinones, triterpenes, and lignans. Furthermore, compounds such as quercetin, gallates, luteolin, betulinic acid, aloe emodin, indigo, and quinomethyl triterpenoids can inhibit viral proteases [22,33].

The profound impact of developing vaccines and plasma therapy is evident in preventing and treating COVID-19 and influenza. Nevertheless, there is still a shortage of medical capacity in all countries. Given this, and regarding the low bioavailability of several medicinal plants and bioactive compounds to humans, nanocarriers like nanoparticles, alcohol solutes, and nanoliposomes can be loaded with these natural agents and effectively deliver them to specific parts of the body to maximize their bioavailability and minimize their already minimal side effects [34,35].

Due to the vastness of bioactive compounds present in medicinal plants, this study aimed to review the role of these plants in COVID-19 and influenza. Moreover, we intended to investigate the potential delivery systems for new antiviral therapies.

## 2. Results

From the sixteen articles selected, 1828 participants were included, 905 with COVID-19, 328 with influenza, 114 exposed adults, and 481 healthy volunteers. Five studies reported only the gender of the participants who completed the study. The age range was 4 years old and above.

Of the 16 articles (five from the USA, one from India, two from China, one from Japan, one from Korea, five from Iran, and one from Belgium, The Netherlands, and France), nine were a randomized, double-blind, placebo-controlled clinical trial, one control clinical trial, two open-label randomized clinical trials, one randomized clinical trial, one triple-blind randomized placebo-controlled clinical trial, one single-blind randomized placebo-controlled clinical trial, and one randomized, open-label, proof-of-concept trial.

One study used *Viola odorata* L. aqueous extract (violet syrup) [36]; one study used pomegranate juice added to SUMAC (a composition of tannins, flavonoids, anthocyanins,

isoflavones, terpenoids, and diterpenes) [37]; one study used Covexir (*Ferula foetida* oleo-gum) [38]; one study used Zufa syrup (a composition of *Nepeta bracteata*, *Ziziphus jujube*, *Glycyrrhizaglabra*, *Ficuscarica*, *Cordia myxa*, *Papaver somniferum*, Fennel, *Adiantumcapillus veneris*, *Viola*, Viper's-buglosses, Lavender, and Iris) [39]; one study used sachets of *Matricaria chamomilla* L., *Zataria multiflora* Boiss., *Glycyrrhiza glabra* L., *Ziziphus jujuba* Mill., *Ficus carica* L., *Urtica dioica* L., *Althaea officinalis* L., *Nepeta bracteata* Benth, and capsules of *Rheum palmatum* L. rizhome, *Glycyrrhiza glabra* root, *Punica granatum* L. fruit peel, *Rheum palmatum*, and *Nigella sativa* L. [40]; one study used giloy, swasari ras, ashwagandha, and tulsi ghanvati [41]; one used a multicomponent over-the-counter formulation [42]; one used lacto-wolfberry [43]; one used aged garlic extract powder [44]; one used ephedra herb, apricot kernel, cinnamon bark and glycyrrhiza root [45]; one used ginseng [46]; two used broccoli sprout decoction [47,48]; one used a *Chima qingwen* decoction [49]; one used elderberry extract orally [50]; and one use posaconazole [51]. The administered doses ranged from 300 mg to 200 g per day, and the intervention period ranged from 4 days to 20 weeks. Two studies used live attenuated influenza virus [47,48].

Studies have shown that medicinal plants (in the different formulations presented) help with faster and more simultaneous recovery, and reduce the risk of viral spread. They reduce inflammatory markers and decrease the incidence and severity of flu and COVID-19. Furthermore, these interventions have protected individuals from contracting acute respiratory illnesses and may tend to reduce the duration and symptom scores of these illnesses. Reported side effects were elevated serum aminotransferase level after treatment, dry mouth, constipation, rash, and sour taste. Patients also had significantly higher serum influenza-specific immunoglobulin G levels post-vaccination and a higher conversion rate (Table 1). Several of the studies we examined did not include control patients for comparison purposes. Table 2 shows the description of the biases observed in the included studies.

**Table 1.** Clinical trials showing the effects of medical plants on COVID-19 and influenza.

| Reference | Local | Model and Patients | Intervention | Outcomes | Adverse Effects |
|---|---|---|---|---|---|
| COVID-19 | | | | | |
| [36] | Iran | Placebo-controlled randomized, double-blind clinical trial with 108 COVID-19 outpatients, 55 male; 53 female; ≥30 y. | 10 mL of violet syrup (*Viola odorata* L. aqueous extract) thrice daily for 7 days. | Patients who received violet syrup ameliorated faster and had lower mean severity scores regarding cough, myalgia, headache, and diarrhea. This shows that violet syrup effectively controlled prevalent COVID-19 manifestations. | No serious adverse events were reported during the trial. |
| [37] | Iran | Placebo-controlled randomized, single-blind clinical trial with 178 COVID-19 outpatients, 120 male; 58 female; ≤60 y. | 200 mL of pomegranate juice thrice daily and 1.5 g of SUMAC (composed of tannins, flavonoids, anthocyanins, isoflavones, terpenoids, and diterpenes) twice daily. Time of intervention was not elucidated. | Patents who received pomegranate and SUMAC interventions had significant decreases in fever, weakness, cough, chills, smell and taste disorders, diarrhea, shortness of breath, abdominal pain, and vomiting compared with patients who did not undergo this treatment. | Not reported. |
| [38] | Iran | Placebo-controlled randomized, double-blind clinical trial with 50 mild to moderate COVID-19 patients, 34 male; 16 female; ≤80 y. | Covexir (*Ferula foetida* oleo-gum) twice daily for 7 days. | Covexir inhibited cough and diminished the severity of anorexia, sense of taste, anosmia, and myalgia between the intervention and placebo groups. | No adverse events were reported during the trial. |
| [39] | Iran | Placebo-controlled randomized triple-blind clinical trial with 116 mild to moderate COVID-19, 57 men; 59 females; 20–70 y. | 7.5 mL of Zufa syrup (*Nepeta bracteata*, *Ziziphus jujube*, *Glycyrrhizaglabra*, *Ficuscarica*, *Cordia myxa*, *Papaver somniferum*, Fennel, *Adiantumcapillus veneris*, *Viola*, Viper's-buglosses, Lavender, and Iris) every 4 h for 10 days. | There were no significant differences between placebo and intervention groups in cough, dyspnea, anxiety, anorexia, insomnia, myalgia, and oxygen saturation decline occurrence. | No serious adverse effects were reported during the trial. |

**Table 1.** *Cont.*

| Reference | Local | Model and Patients | Intervention | Outcomes | Adverse Effects |
|---|---|---|---|---|---|
| [40] | Iran | Multicenter open-labeled, randomized, controlled clinical trial with 358 mild to moderate COVID-19 patients, 197 male; 161 female; ≤75 y. | Patients received a polyherbal decoction (one sachet of the following per day respecting the order: *Matricaria chamomilla* L., *Zataria multiflora* Boiss., *Glycyrrhiza glabra* L., *Ziziphus jujuba* Mill., *Ficus carica* L., *Urtica dioica* L., *Althaea officinalis* L., and *Nepeta bracteata* Benth) every 8 h and two herbal capsules (*Rheum palmatum* L. rizhome, *Glycyrrhiza glabra* root, *Punica granatum* L. fruit peel, and *Rheum palmatum* for capsule one and *Nigella sativa* L. for capsule two) every 12 h for 7 days. | Patients who received the intervention had significantly lower dyspnea periods, as well as accelerated clinical improvement of dry cough, headache, muscle pain, vertigo chills, fatigue, anorexia, sputum cough, and runny nose. | Gastrointestinal adverse effects like nausea and diarrhea were observed. |
| [41] | India | Placebo-controlled randomized, double-blind pilot clinical trial with 95 patients who had no or mild symptoms of COVID-19 and were positive on RT-PCR, 77 males; 18 females; 15- 80 y. | 1 g of giloy, 2 g of swasari ras, 0.5 g each of ashwagandha, and tulsi ghanvati were given orally to the patients in the treatment group twice per day for 7 days. | Ayurvedic treatment can expedite virological clearance, help rapid recovery, and reduce the risk of viral dissemination and inflammation markers (suggesting a lower severity of SARS-CoV-2 infection in the treated group). | There were no side effects. |
| [42] | USA | Controlled clinical trial with 114 multiply exposed adults, 60 female, 40 males; ≥30 y. | Patients in the treatment group received a daily dose of OTC for 20 weeks, while the control group did not receive any placebo as they refused the study regimen. | Just under 4% of the compliant test group presented flu-like symptoms, but none of the test group was COVID-positive; whereas 20% of the non-compliant control group presented flu-like symptoms, three-quarters of whom (15% of the overall control group) were COVID-positive. | Not reported. |
| **Influenza** | | | | | |
| [43] | China | Randomized, double-blinded, placebo-controlled study with 150 healthy community-dwelling Chinese elderly, 75 male and 75 female; 65–70 y. | The treated group received a single-dose sachet containing 13.7 g/day lacto-wolfberry (wolfberry fruit (530 mg/g), bovine skimmed milk (290 mg/g), and maltodextrin, 180 mg/g). The placebo group received the same sachet but with a skimmed bovine formulation of milk, maltodextrin, sucrose, and colorants/92 d. | The treated subjects showed significantly higher immunoglobulin G levels post-vaccination and in seroconversion (between days 30 and 90, compared with the placebo group). | No serious adverse effects were reported during the trial. |
| [44] | USA | Randomized, double-blind, placebo-controlled with 120 healthy men (55) and women (65), 21–50 y with BMI 18–30 kg/m². | Each patient consumed four capsules/day of either AGE (2.56 g per day) or a placebo for 90 days. | The use of aged garlic extract enhanced immune cell function (possibly responsible for the reduction of cold and flu severity.) | Not reported. |
| [45] | Japan | Open-labeled, randomized controlled trial with 33 patients, 14 male; 14 female; 20–64 y, presented within 48 h of onset of flu symptoms, including fever, and were positive by quick diagnostic test kit for influenza virus antigens from nasal swabs. | Patients were randomized into three groups to receive Maoto orally at 2.5 g TID, or Oseltamivir orally 75 mg BID, or Zanamivir by inhalation of 20 mg BID for 5 days. | The administration of oral Maoto granules to healthy adults with seasonal influenza was well tolerated and associated with equivalent clinical and virological efficacy to neuraminidase inhibitors. | One patient in the Maoto group and one in the Oseltamivir group showed a mildly elevated serum aminotransferase level after treatment. |
| [46] | Korea | Randomized, double-blinded, placebo-controlled trial in 100 healthy volunteers, 38 male; 62 female, 30–70 y. | The treatment group received concentrated red ginseng 1.0 g three times a day (3.0 g/day)/12 w. Placebo was similar in taste and appearance but with no principal ingredients. | KRG is effective in protecting subjects against ARI and may decrease the duration and scores of ARI symptoms. | There were no specific clinical and laboratory side effects. |
| [48] | USA | A randomized, double-blind, placebo-controlled trial with 69 healthy young adult smokers and nonsmokers, 26 female; 25 male; 18–40 y. | Smokers and nonsmokers ingested one daily dose 200 g of BSH or placebo (ASH) for 4 days. On day 0 they received a standard vaccine dose of LAIV intranasally. | In smokers, short-term intake of BSH appears to significantly reduce some markers of inflammation, such as IL-6, and reduce the amount of the influenza virus. | No patients reported intolerable taste or side effects. |
| [47] | USA | Randomized, double-blinded, placebo-controlled study with 42 healthy volunteers, 19 female; 10 male; 25–28 y. | Subjects received BSH or placebo (ASH) for 4 consecutive days. A daily portion of BSH shake was about 200 g. On day 0 they received a standard vaccine dose of LAIV intranasally. | BSH increases virus-induced peripheral blood NK cell granzyme B production, an effect that may be important for enhanced antiviral defense responses. | No subject reported intolerable taste or side effects. |

**Table 1.** *Cont.*

| Reference | Local | Model and Patients | Intervention | Outcomes | Adverse Effects |
|---|---|---|---|---|---|
| [49] | China | Randomized clinical trial with 120 subjects who have mild influenza A (H1N1). Including 62 males and 58 females, 14–65 y. | The treated group received chima qingwen decoction two times a day for 5 days (children received half the dose). The antiviral group took oral Oseltamivir every 75 mg (50 mg children), two times a day, one course of 5 days. | The overall effective rate was 93.3%. A combination of therapy (Chinese and Western medicine) is effective for mild cases of influenza A (H1N1). | No adverse effects occurred. |
| [50] | USA | Randomized, double-blind, placebo-controlled trial with eighty-seven patients, ≥4, with less than 48 h of at least two moderate-severity symptoms of influenza and positive polymerase chain reaction influenza test, 49 males; 38 females. | Participants from age 5 to 12 y received a placebo or 15 mL (5.7 g) elderberry extract orally 2× d for 5 d; those > 12 years received 15 mL 4× d for 5 d. Patients were permitted to choose to also receive the standard dosage of Oseltamivir. | No evidence was found that the elderberry benefits the duration or severity of the flu. | Dry mouth, constipation, rash, and bad taste. There were no significant differences between the elderberry and placebo. |
| [51] | Belgium, The Netherlands, and France | Randomized, open-label, proof-of-concept trial with 88 critically ill influenza patients, 41 male; 32 female, ≥18 y. | Participants submitted to the prophylaxis arm received the first dose of POS prophylaxis within 48 h of admission to the ICU, starting with a loading dose of 300 mg 2× d on day 1, followed by a 1× d of 300 mg from day 2 onwards for 7 days. The other group received the standard of care only. | The higher-than-expected incidence of early IAPA precluded any definitive conclusions about POS prophylaxis. After 48 h, still, 11% of patients developed IAPA. | Not reported. |

AGE: aged garlic extract powder; ARI: acute respiratory illness; ASH: alfalfa sprout homogenate; ashwagandha: (Withania somnifera); BMI: Body Mass Index; BSH: broccoli sprout homogenates; chima qingwen decoction: Bupleurum 20 g + Scutellaria baicalensis 10 g + Pinellia ternate 10 g + Radix pseudostellariae 10 g + ginger 10 g, jujube 10 g + licorice 6 g, on this basis, adding gypsum, honeysuckle, forsythia, Folium isatidis; COVID-19: coronavirus disease; D: day; G: gram; giloy: Giloy Ghanvati (Tinospora cordifolia); H: hour; H1N1: influenza A virus subtype H1N1; IAPA: influenza-associated pulmonary aspergillosis; ICU: intensive care unit; IL-6: interleukin-6; Kg: kilogram; KRG: Korean red ginseng; LAIV: live attenuated influenza virus; m2: square meter; Maoto: multicomponent formulation extracted from four plants: ephedra herb, apricot kernel, cinnamon bark, and glycyrrhiza root; Mg: milligram; mL: milliliter; NK cell: natural killer cell; Oseltamivir: Tamiflu; OTC: multicomponent over-the-counter "core formulation" containing 25 mg zinc; 10 drops of Quina; 400 mg quercetin; 1000 mg vitamin C; 1000 IU (25 mg) vitamin D3; 400 IU Vitamin E; and 500 mg l-lysine; POS: posaconazole; Roche; Zanamivir: (Relenza; GlaxoSmithKline); RT-PCR: reverse transcription polymerase chain reaction; SARS-CoV-2: severe acute respiratory syndrome coronavirus 2; swasari ras: (traditional herb-mineral formulation); TID; ter in die; tulsi ghanvati: (Ocimum sanctum); Xd: times a day; Y: years

**Table 2.** Descriptive table of the biases of the included randomized clinical trials.

| Study | Question Focus | Appropriate Randomization | Allocation Blinding | Double-Blind | Losses (<20%) | Prognostics or Demographic Characteristics | Outcomes | Intention to Treat Analysis | Sample Calculation | Adequate Follow-Up |
|---|---|---|---|---|---|---|---|---|---|---|
| [36] | Yes | Yes | Yes | Yes | Yes | Yes | Yes | Yes | Yes | Yes |
| [37] | Yes | No | Yes | No | Yes | No | Yes | No | Yes | NR |
| [38] | Yes | No | Yes | Yes | No | Yes | Yes | No | No | Yes |
| [39] | Yes | Yes | Yes | Yes | Yes | Yes | Yes | No | Yes | Yes |
| [40] | Yes | Yes | No | No | Yes | Yes | Yes | Yes | Yes | Yes |
| [41] | Yes | Yes | Yes | Yes | Yes | Yes | Yes | N | No | Yes |
| [42] | Yes | No | No | No | Yes | NR | Yes | No | Yes | Yes |
| [43] | Yes | Yes | Yes | Yes | Yes | NR | Yes | Yes | Yes | Yes |
| [44] | Yes | Yes | Yes | Yes | Yes | NR | Yes | Yes | NR | Yes |
| [45] | Yes | Yes | Yes | No | Yes | Yes | Yes | Yes | Yes | Yes |
| [46] | Yes | Yes | Yes | Yes | Yes | Yes | Yes | Yes | NR | Yes |
| [48] | Yes | Yes | Yes | Yes | No | Yes | Yes | Yes | Yes | Yes |
| [47] | Yes | Yes | Yes | Yes | No | Yes | Yes | Yes | Yes | Yes |
| [49] | Yes | NR | NR | NR | NR | Yes | Yes | NR | Yes | Yes |
| [50] | Yes | Yes | Yes | Yes | Yes | Yes | Yes | Yes | Yes | Yes |
| [51] | Yes | Yes | Yes | No | Yes | Yes | Yes | Yes | No | Yes |

NR, not reported.

### 3. Discussion

*3.1. COVID-19*

Severe acute respiratory syndrome (SARS) and coronavirus-caused Middle East respiratory syndrome (MERS) are coronaviruses that could spread globally in the next few years. Coronaviridae is the family of viruses that comprises SARS-CoV-2, the third coronavirus strain that caused a pandemic. Although the SARS-CoV-2 origin is unknown, it is notorious that this virus comprises a single-stranded positive-sense RNA genome. Furthermore, this virus is giant and enveloped, scaling from 60 nm to 140 nm in diameter and having spikes of 9 to 12 nm. The spikes of SARS-CoV-2 are responsible for giving the virions the aspect of a solar corona. The genome of this coronavirus includes four different structural proteins that are important for its infectiousness: nucleocapsid (N protein), membrane (M protein), envelope (E protein), and spike (S protein). Transmission occurs principally via face-to-face contact or contaminated surfaces. Respiratory droplets are the main ones responsible for the spread of the virus, but aerosol spread can also be present. Face-to-face contact with contaminated respiratory droplets is the leading cause of transmission, mainly because asymptomatic people also spread the virus. Respiratory-contaminated droplets can transfer the virus from one infected human to another, even when face-to-face contact with ocular surfaces occurs. The infection causes common symptoms and signals through the infected people, such as fever, dry cough, shortness of breath, nausea, fatigue, myalgia, vomiting, diarrhea, headache, weakness, rhinorrhea, and anosmia or ageusia [52–55].

The pathophysiology of a SARS-CoV-2 infection is triggered when the virus first enters the nasal respiratory tract's epithelial cells. In these nasal cells, the virus multiplies and starts to infect the lower respiratory areas using the angiotensin-converting enzyme receptor 2 (ACE2). In summary, the nasal epithelial cells serve for the virions to replicate using the RNA genome. Before large replications, these virions are released to infect even lower areas of the respiratory tract, reaching the alveolar zone of the lungs. COVID-19 is considered a multi-organ disease due to the presence of ACE2 receptors in many organs beyond the respiratory epithelium: the brain, kidney, pancreas, cardiovascular endothelium, liver, and bowel are all sites of the human body that exhibit ACE2 receptors. Pulmonary involvement is the center of coronavirus disease. The virus enters the pulmonary alveolar cells via endocytosis by binding the S protein to the ACE2 receptor, which includes the activation of the S protein by the type 2 transmembrane serine protease (TMPRSS2) and the cleavage of the ACE2 receptor. In the interior of the pulmonary cells, the virions release the RNA genome and start replicating. This multiplication is responsible for forming many new virions, which are liberated and infect new cells. In the meantime, immune cells start an inflammatory response against the presence of SARS-CoV-2 in the lungs. Lymphocytes, monocytes, macrophages, and neutrophils enhance cytokine release, which causes acute injuries to the pulmonary organs.

In summary, the vasculature of the lungs becomes very porous. Combined with the inflammatory response, the vasculature changes lead to pulmonary edema, pulmonary ischemia, activation of intravascular coagulation, respiratory failure (hypoxia), and progressive lung damage. Hyaline membrane formation is observed, and the pulmonary findings can be characteristic of the early-phase acute respiratory distress syndrome. In inflammation, oxidative stress or redox imbalance can also be associated with the severe acute respiratory disease caused by SARS-CoV-2. Oxidative phenomena are highly associated with inflammatory ambiances. The presence of reactive oxygen species and the decreased activities of antioxidant mechanisms are essential for viral replications [52–54,56].

COVID-19 associates with hyperinflammatory states. If the immune recruited cells end the infection in the pulmonary tissues, the disease recedes. However, the innate immune response can be dysregulated, and a hyperinflammatory syndrome starts. This inflammatory dysregulation is called a cytokine storm. The cytokines associated with this abnormal release of inflammatory cytokines syndrome are interleukin 6 (IL-6), IL-1, IL-10, IL-2, tumor necrosis factor alfa (TNF-$\alpha$), and interferon-gamma (IFN-$\gamma$). It is known

that the measurement of pro-inflammatory markers is crucial to premeditate the severity and mortality associated with COVID-19. The prognostic value of the cytokine storm is helpful because no survivors of the SARS-CoV-2 infection exhibit high inflammatory responses. The cytokine storm might derive primarily from the vasculature changes, leading to the activation of complement proteins and the accumulation of inflammatory cells in the pulmonary tissue, such as pro-inflammatory monocytes, macrophages, and neutrophils. People affected by high inflammation are also the group of patients with the higher prevalence of acute respiratory distress syndrome. The scenario of the cytokine storm is represented mainly by Th1 cell immune responses and M1 macrophage-polarized domains. SARS-CoV-2 is a cytopathic virus, which means that, in the process of viral replication, the epithelial cells of the lungs are led to death due to pyroptosis. This type of programmed death is pro-inflammatory and can stimulate the hyperinflammatory state that encodes the cytokine storm principally because, in pyroptosis, the cells release high amounts of IL-1β. In addition, pyroptosis also stimulates the release of pathogen-associated molecular patterns from the infected cells. Other epithelial cells of the lungs and resident alveolar macrophages identify these patterns through the activity of pattern-recognition receptors. The stimulatory effects of these patterns and IL-1β promote the release of more pro-inflammatory cytokines. These cytokines activate more immune cells in the lungs and more inflammatory cytokines are released, contributing to the destruction of the lungs' parenchyma. Because of these events at the site of the SARS-CoV-2 infection, an inflammatory feedback loop begins, and systemic effects such as sepsis and multi-organ failure occur [57–62].

The COVID-19 pandemic brought about a reorganization of the healthcare landscape. The immense pressure to produce billions of vaccines within a limited timeframe significantly strained the vaccine production chain. Consequently, these chains encountered challenges in meeting the high demand, leading to disruptions and delays in production [63]. Furthermore, there has been substantial growth in molecular biology research. A notable example is the advancements in understanding the role of cell surface protein HSPA5/BiP/GRP78 in both cancers and COVID-19. HSPA5, highly expressed in malignant tumors, seems pivotal in the invasion/attack of SARS-CoV-2 in cancer patients through tumor tissues. Consequently, there has been a growing discussion about the therapeutic and prognostic implications of targeting HSPA5 in both cancer and COVID-19. Exploring HSPA5 expression through natural products may hold promising clinical significance for future endeavors to combat COVID-19 and cancer [64].

Figure 1 represents the main aspects of the SARS-CoV-2 infection and the pathophysiological routes of the disease.

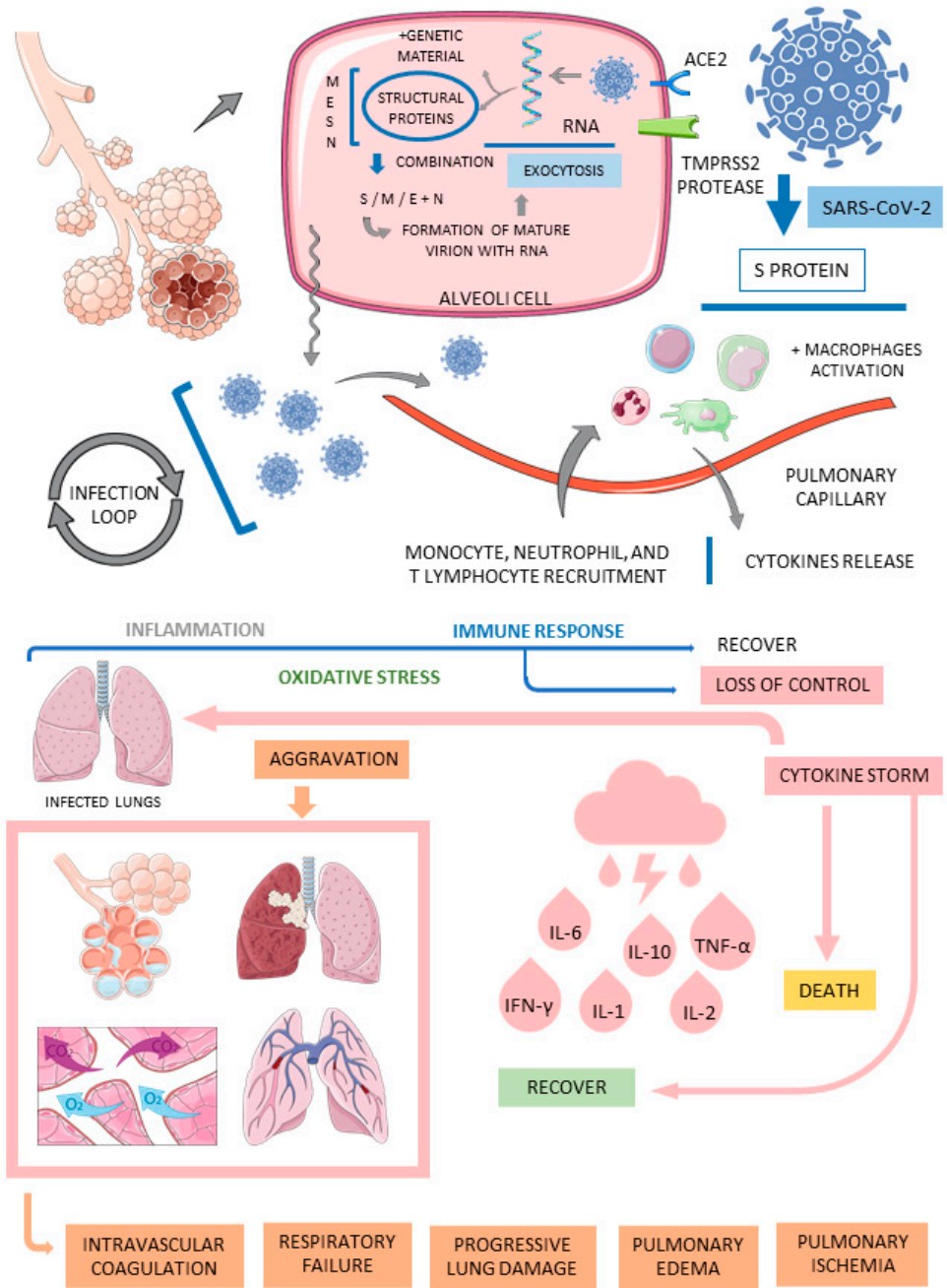

**Figure 1.** Aspects of the SARS-CoV-2 infection and possible routes of the disease. M: membrane, N: nucleocapsid, E: envelope, S: spike, ACE2: angiotensin-converting enzyme receptor 2, TMPRSS2: type 2 transmembrane serine protease, IL-6: interleukin 6, IL-10: interleukin 10, IL-1: interleukin 1, IL-2: interleukin 2, TNF-α: factor tumor necrosis alfa, IFN-γ: interferon-gamma, +: plus.

### 3.2. Influenza

Influenza viruses are responsible for causing acute respiratory disease in many mammals, including humans. These viruses affect the respiratory system causing infection, principally by direct viral infection and damage derived from the immunological response. The characteristics of the disease depend on the subtype of the influenza viruses. There are four different subtypes of influenza: influenza A, B, C, and D. Although only subtypes A and B cause seasonal flu, in the world, the influenza epidemic occurs all the time. These viruses are considered zoonotic, principally because the origins are related to reservoirs in bats and a variable of aquatic birds. Influenza viruses present an enormous variety due to the ability of the viruses to change very simply. The primary site of infections is the

respiratory epithelium; however, often some immune cells can be infected and initiate viral replication. Influenza viruses present a negative-sense RNA genome that progressively accumulates mutations, principally due to this genomic lack of proofreading mechanisms. These alterations that happen continuously in the RNA of the viruses are the main factor responsible for influenza pandemics. The transmission is related to respiratory particles. Contaminated infectious respiratory particles are created when infected people sneeze or cough, and the particles contact healthy people via inhalation. Moreover, seasonal influenza correlates with an incubation period of 24 to 48 h; however, infected people transmit the viruses before (one to two days) and after (five to seven days) the onset of the symptoms. Common symptoms of the infection are cough, myalgia, chills, fever, and malaise. Although the symptoms correspond to an uncomplicated respiratory tract infection, influenza infection can bring other, worse outcomes. The complications are mainly cardiovascular, musculoskeletal, neurologic, and pulmonary [65–68].

The influenza infection's pathophysiology derives from the viruses' actions in infecting the upper and lower respiratory tracts. For these reasons, all cells in the respiratory system epithelium can be infected, including the nasal epithelium (the highest) and alveoli epithelium (the lowest). Inflammation is the primary pathophysiology mechanism of influenza infection. The immune responses can be related to worsening the commitment of the disease to the respiratory tract, principally due to inflammation. Influenza A is an influenza-like virus that mostly overburdens global health. Influenza A is mainly related to pandemic outbreaks when novel subtypes arrive, making this type of influenza virus the most significant preoccupation of the world's public health experts. This happens principally because, with influenza A, new viruses can emerge from animals. Influenza B and C are only related to causing diseases related to epidemic proportions. Two influenza A viruses affect people nowadays (H1N1 and H3N2). Influenza A virus subtypes are associated with the glycoproteins that can appear on the virus's surfaces. Hemagglutinin (16 to 18 subtypes) and neuraminidase (nine to 11) can interfere with infection rates and mortality. Hemagglutinin anchors the influenza virion to the human cell surface, and neuraminidase causes digestion of the host secretions, allowing the release of the viral particles from the infected cells of the host. When the alveolar epithelium of the lungs is affected, the host is at a high risk of developing severe disease, primarily due to the exposure of the viruses to the endothelial cells of the lungs. This phenomenon occurs when the influenza infection mediates the destruction of alveoli's structures related to gas exchange in the respiratory epithelium [67,69–71].

Influenza is a health concern that presents a significant opportunity for technological advancements each year, primarily due to its seasonal nature. Recently, the importance of vaccination has been extensively demonstrated, proving to be the most effective defense against various infections, including SARS-CoV-2. Parenteral vaccination, which involves injecting vaccines into the body, is the most commonly used method for immunization against systemic and respiratory diseases and central nervous system disorders. It activates T and B cells, triggering a comprehensive immune response. However, mucosal vaccines, such as nasal vaccines, offer an additional advantage by stimulating immune cells in the mucosal tissues of the upper and lower respiratory tracts [72]. Strategies that imitate or hinder the function of neuraminidases present intriguing possibilities for treating viral and bacterial infections. Neuraminidases, also known as sialidases, are a class of enzymes that regulate the activity of sialic acids, which are known to have crucial roles in various biological processes and pathological conditions. These enzymes can be found in mammals, as well as in viruses and bacteria. This inherently multidisciplinary field encompasses structural biology, biochemistry, physiology, and the study of host-pathogen interactions. It offers exciting research opportunities that can enhance our understanding of the mechanisms involved in virus-bacteria co-infections and their impact on exacerbating respiratory diseases, particularly in the context of pre-existing pathological conditions [73].

Figure 2 represents the pathophysiological events of influenza infection and the potential medicinal plants' inhibitory effects on pathological pathways.

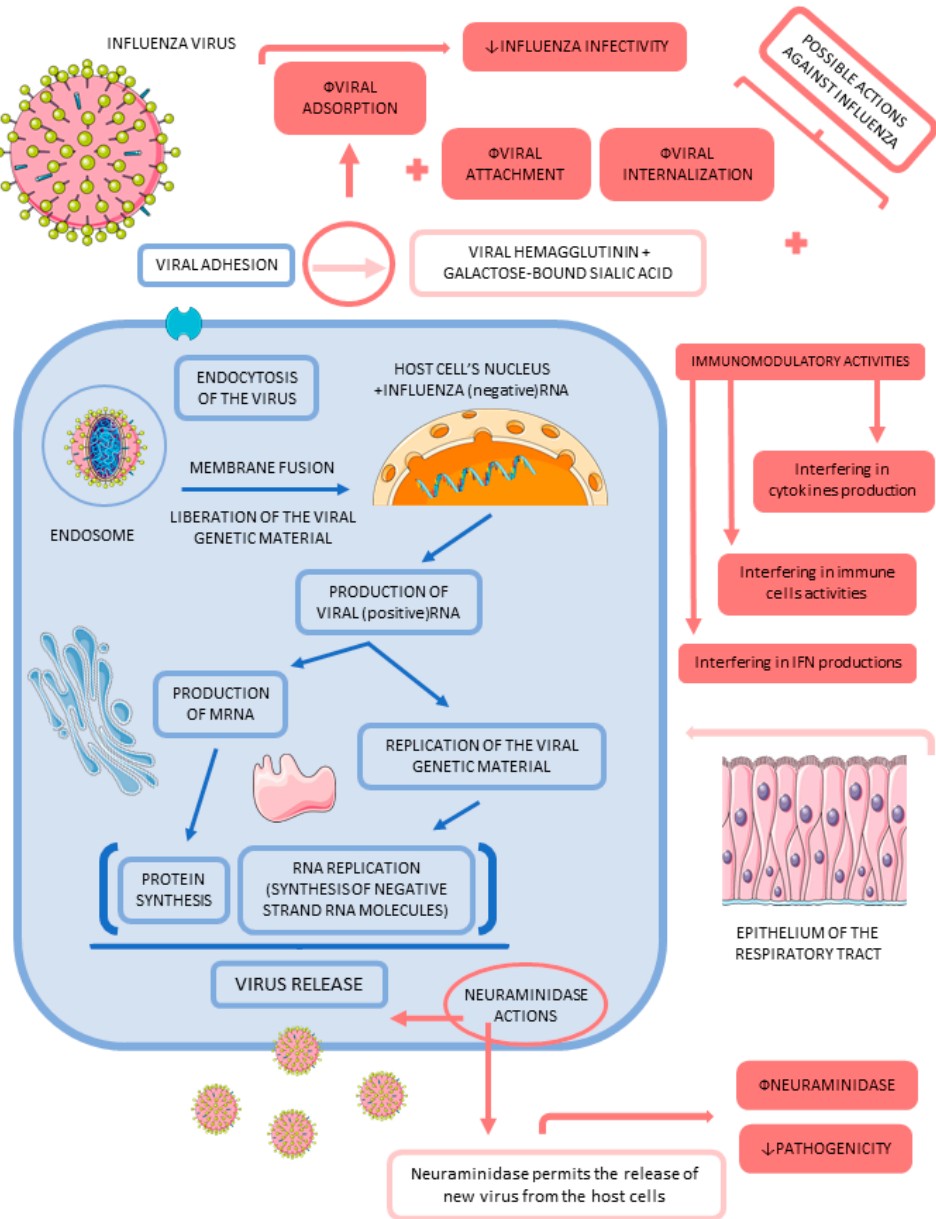

**Figure 2.** Features of influenza infection pathogenesis and potential pathways of inhibition. ↓: decrease, φ: inhibition.

### 3.3. The Role of Medicinal Plants on COVID-19

A genuinely potent antiviral agent against COVID-19 is lacking. Although much research has been done for effective vaccines and drug agents against SARS-CoV-2, effective therapies have not yet been found. For these reasons, preventive and supportive therapeutics are usually used to manage a patient infected by SARS-CoV-2, mainly to control complications or avoid organ damage. Plants and herbs were previously shown to be natural substances corresponding to antiviral activities and producing anti-inflammatory and antioxidant actions. Common antivirals are associated with limited efficacy rates and can promote serious side effects.

Medicinal plants present diverse bioactive compounds that strongly correlate with the treatment and prevention of COVID-19. Mostly in the form of mixtures, medicinal plants and plants metabolites are rich in antiviral compounds and can integrate plant-based therapies into the fight against SARS-CoV-2 infection [74–79]. Moreover, natural compounds are more tolerable than typical medications and inexpensive.

Antiviral phytocompounds act through several mechanisms against SARS-CoV-2, principally interfering in COVID-19 pathophysiology. Medicinal plants often present many different bioactive compounds that have antiviral effects and promote anti-inflammatory and immunomodulatory effects. This combination of activities can augment the efficacy of using a specific plant to treat a viral infection [74,80,81].

The first line of defense promoted by natural compounds against SARS-CoV-2 is related to the entrance into the cells, inhibiting the interaction between the viral spike protein and the receptor that interacts with this structural component of the virus. Spike proteins and ACE2 receptors interact via the spike glycoprotein receptor-binding domain. This domain recognizes the ACE2 receptor, leading to the viral particles' internalization into the host cells. The internalization of the viral particles permits the genome of the SARS-CoV-2 to penetrate the host cells. Due to these facts, blockers of ACE2 receptors can be effective against the pathogenesis of COVID-19 disease. Furthermore, the virus-ACE2 interaction depends on a protease called TMPRSS2. This molecule corresponds to a transmembrane serine protease responsible for the cleavage of the protein spike and the ACE2 receptor, an essential phenomenon during viral internalization. Due to these reasons, TMPRSS2 inhibitors can also be necessary in combating SARS-CoV-2. Medicinal plants can exhibit several bioactive compounds associated with ACE2 receptors and TMPRSS2 protease inhibitions. Emodin and caffeic acid are molecules that promote activities that inhibit ACE2 receptors. Kaempferol, luteolin, sulforaphane, quercetin, and cryptotanshinone promote activities that inhibit TMPRSS2 proteases. The inhibitory actions of medicinal plants' compounds on ACE2 and TMPRSS2 can correspond to a necessary adjuvant treatment against COVID-19. The effective blockage of the virus' entry into the host cells can significantly inhibit infection [74,77,78,80].

Interfering in SARS-CoV-2 replication integrates with inhibitory effects against the pathogenic routes of COVID-19 infection. The causal virus of COVID-19 necessitates the replication of enzymes. The proteolytic activities of two enzymes are essential to replicating polyproteins during the viral maturation in the host cells. These two proteins are chymotrypsin-like protease [3CL(pro)] and papain-like proteinase [PL(pro)]. PL(pro) is a non-structural protein of SARS-CoV-2 that plays roles in the cleavage process of viral polyproteins into action-active proteins. In addition, PL(pro) corresponds to an antagonist of the innate immunological system, mainly targeting interferon production and the signaling pathways of nuclear factor-kappa B (NF-kB). In turn, 3CL(pro) is also a nonstructural protein encoded by the SARS-CoV-2 genetic material that exerts effects on the viral replication processing polyproteins of the virus. Amentoflavone, herbacetin, pectolinarin, rhoifolin, dihydrotanshinone, and gallocatechin gallate are bioactive compounds of different medicinal plants that exert potential inhibitors of SARS-CoV-2 3CL(pro) protease. Dieckol, hirsutenone, tomentin E, and psoralidin can exert possible effects inhibiting SARS-CoV-2 PL(pro) protease [74,77,78,80,82–86].

Another phytotherapy against COVID-19 can correspond to helicase inhibitors. SARS-CoV-2 uses helicases known as NTPase to replicate the viral genome and the transcript and translate this genomic material. There are two bioactive compounds referred to that inhibit the activities of SARS-CoV-2 helicase: myricetin and scutellarein. These two compounds inhibit the helicase by inhibiting the ATPase activity associated with the correct function of the SARS-CoV-2 helicase non-structural proteins. Helicase is a protein related to synthesizing variable parts of the virus, from viral structural proteins to viral enzymes. The inhibition of helicase may discourage the assembly of the mature SARS-CoV-2 virions at the final stage of the replication process. Therefore, the inhibition of helicase protein might be considered in treating COVID-19 infection. SARS-CoV-2 RNA-dependent RNA polymerase (RdRp) inhibitors have also been evaluated as potent agents against COVID-19. This enzyme is essential, principally due to its synthesis of SARS-CoV-2 sense and antisense RNAs associated with viral replication. In Table 3, it is possible to find some relevant bioactive compounds that can exert anti-COVID-19 activity [87–91].

**Table 3.** Some main bioactive compounds that can exert antiviral effects.

| Bioactive Compound | Molecular Structure | Antiviral Function (s) | References |
|---|---|---|---|
| Bioactive Compounds of Medicinal Plants That Mainly Affect SARS-CoV-2 | | | |
| Casticin |  | - Immunomodulatory and anti-inflammatory actions on lungs. | [82] |
| Emodin |  | - Blockage of the interaction between SARS-CoV-2 S protein and ACE2 receptor;<br>- Inhibition of SARS-CoV-2 S protein. | [77,78,89] |
| Dieckol |  | - Inhibition of SARS-CoV-2 PL(pro) protease. | [77] |
| Curcumin |  | - Probable inhibition of SARS-CoV-2 S protein;<br>- Probable inhibition of ACE2;<br>- ↓ Pro-inflammatory cytokines;<br>- ↑ Toxicity of NK cells. | [78,92] |
| Myricetin |  | - Inhibition of SARS-CoV-2 helicase affecting ATPase activity. | [77,89] |
| Hirsutenone |  | - Inhibition of SARS-CoV-2 PL(pro) protease. | [77] |
| Scutellarein |  | - Inhibition of SARS-CoV-2 helicase affecting ATPase activity. | [77,89] |
| Quercetin |  | - ↓ Pro-inflammatory cytokines;<br>- ↑ Expression of IL-4 and IL-5 cytokines;<br>- Inhibition of COVID-19 $M_{pro}$ and SARS-CoV-3CL(pro) activity;<br>- Probable inhibition of TMPRSS2 and ACE2; | [77,78,92,93] |

**Table 3.** *Cont.*

| Bioactive Compound | Molecular Structure | Antiviral Function (s) | References |
|---|---|---|---|
| Sulforaphane |  | - Downregulation of TMPRSS2 expression. | [77] |
| Resveratrol |  | - Inhibitory effects on TNF-α and NF-kB. | [92] |
| Ginsenoside |  | - Stimulatory effects on the proliferation of T-helper cells. | [92] |
| Kaempferol |  | - Probable inhibition of TMPRSS2;<br>- Inhibition of the 3a ion channel of the coronavirus. | [77,93] |
| Caffeic Acid |  | - Inhibition of ACE2;<br>- Inhibition of viral attachment. | [77,78] |
| Desmethoxyreserpine | --- | - Affects virus entry in the cells and virus multiplication. | [78,94] |
| Luteolin |  | - Inhibition of TMPRSS2;<br>- Probable inhibition of ACE2 and SARS-CoV-2 S protein. | [77,78,89,93] |
| Betulinic Acid |  | - Affects viral multiplication. | [78] |
| Xanthoangelol E |  | - Inhibition of SARS-CoV-2 3CL(pro) protease. | [95] |
| Hyoscyamine |  | - Viral inhibition;<br>- Bronchodilator. | [82] |
| Cryptotanshinone |  | - Anti-TMPRSS2 activity;<br>- Inhibition of SARS-CoV-2 PL(pro) and SARS-CoV-2 3CL(pro). | [77,78] |
| Allicin |  | - Affects viral replication;<br>- Stimulatory effects of the T lymphocyte proliferation. | [82,92] |
| Dihydrotanshinone-1 |  | - Affects virus entry in the cells;<br>- Inhibition of SARS-CoV-2 3CL(pro) protease. | [77,78,94] |

**Table 3.** *Cont.*

| Bioactive Compound | Molecular Structure | Antiviral Function (s) | References |
|---|---|---|---|
| Tartaric Acid | | - Inhibitory effect against the viral DNA production. | [82] |
| Andrographolide | | - Antipyretic via inhibition of IL-1β and IL-1α expressions;<br>- ↓ IL-6 levels. | [80] |
| Amentoflavone | | - Inhibition of SARS-CoV-2 3CL(pro) protease. | [77] |
| Nicotianamine | | - Inhibition of ACE2 receptor. | [89] |
| Glycyrrhizin | | - Inhibition of ACE2 receptor. | [89] |
| Bilobetin | | - Inhibition of SARS-CoV-2 3CL(pro) protease. | [96] |
| Sciadopitysin | | - Inhibition of SARS-CoV-2 3CL(pro) protease. | [96] |
| Bioactive Compounds of Medicinal Plants That Mainly Affect Influenza. | | | |
| Allicin | | - Proteolytic/hemagglutinating activities;<br>- Affects viral replication;<br>- Stimulatory effects of the T lymphocyte proliferation;<br>- Inhibition of nucleoprotein synthesis of the virus and virus polymerase activity. | [82,92,97] |
| Eucalyptol | | - Inactivation activities against influenza A;<br>- Disruption of the virus envelope. | [75,98] |
| Pentagalloylglucose(Polyphenol) | | - Anti-influenza activity;<br>- Reduces virus release and the sites of virus assembly;<br>- Inhibits hemagglutination phenomena induced by influenza. | [74] |

**Table 3.** *Cont.*

| Bioactive Compound | Molecular Structure | Antiviral Function (s) | References |
|---|---|---|---|
| Coumarin |  | - Inhibits proteins related to influenza entry and replication in the cells;<br>- Inhibitory effects against the infectious process. | [74] |
| Ellagic Acid |  | - Protease inhibitor. | [82] |
| Chlorogenic Acid |  | - Inhibition of neuraminidase activity, viral activity, and new virus release from infected cells. | [97,99,100] |
| Isoquercetin |  | - Inhibits replications of influenza A and B. | [97] |
| Luteolin |  | - Inhibits influenza H3N2 neuraminidase;<br>- Reduces virus cytopathic effects;<br>- Inhibition of H1N1 neuraminidase. | [97,101,102] |
| Glycyrrhizin |  | - Probable protection against influenza viruses through the stimulatory effects on IFN-γ production. | [103] |
| Glycyrrhizin Acid |  | - Probable protection against influenza viruses through the inhibitory effects on virus growth;<br>- Inhibitory effects on cytokine production. | [103] |
| Aloin |  | - Inhibition of neuraminidase activity. | [104] |
| Quercetin-3-sophoroside |  | - Inhibits influenza H3N2 neuraminidase. | [97,101] |
| 4-Methoxycinnamaldehyde |  | - Inhibition of viral attachment;<br>- Stimulatory effects on IFN production;<br>- Inhibitory effects on viral internalization. | [104] |
| Punicalagin |  | - Neuraminidase inhibitor;<br>- Blockage of influenza A virus release from infected cells. | [105] |

**Table 3.** *Cont.*

| Bioactive Compound | Molecular Structure | Antiviral Function (s) | References |
|---|---|---|---|
| Agathisflavone | | - Affects influenza replication inhibiting neuraminidase. | [97] |
| Guggulsterone | | - Inhibition of viral absorption. | [104] |
| Chinonin | | - Decreases influenza A infectivity. | [106] |

M$_{pro}$: main protease, S: spike protein of SARS-CoV-2, ACE2: angiotensin-converting enzyme receptor 2, IL-4: interleukin 4, IL-5: interleukin 5, SARS-CoV-2 3CL(pro): 3-chymotrypsin like protease, TMPRSS2: type 2 transmembrane serine protease, SARS-CoV-2 PL(pro): papain-like cysteine protease, TNF-$\alpha$: tumor factor necrosis alfa, NF-kB: nuclear factor kappa B, IFN-$\gamma$: interferon gama, IFN: interferon, IL-6: interleukin 6, IL-1$\beta$: interleukin 1 beta, IL-1$\alpha$: interleukin 1 alfa, NK: natural killers cells, ↓: decrease, ↑: increase.

Figure 3 summarizes the potential pathways to inhibit COVID-19 infection using medicinal plants.

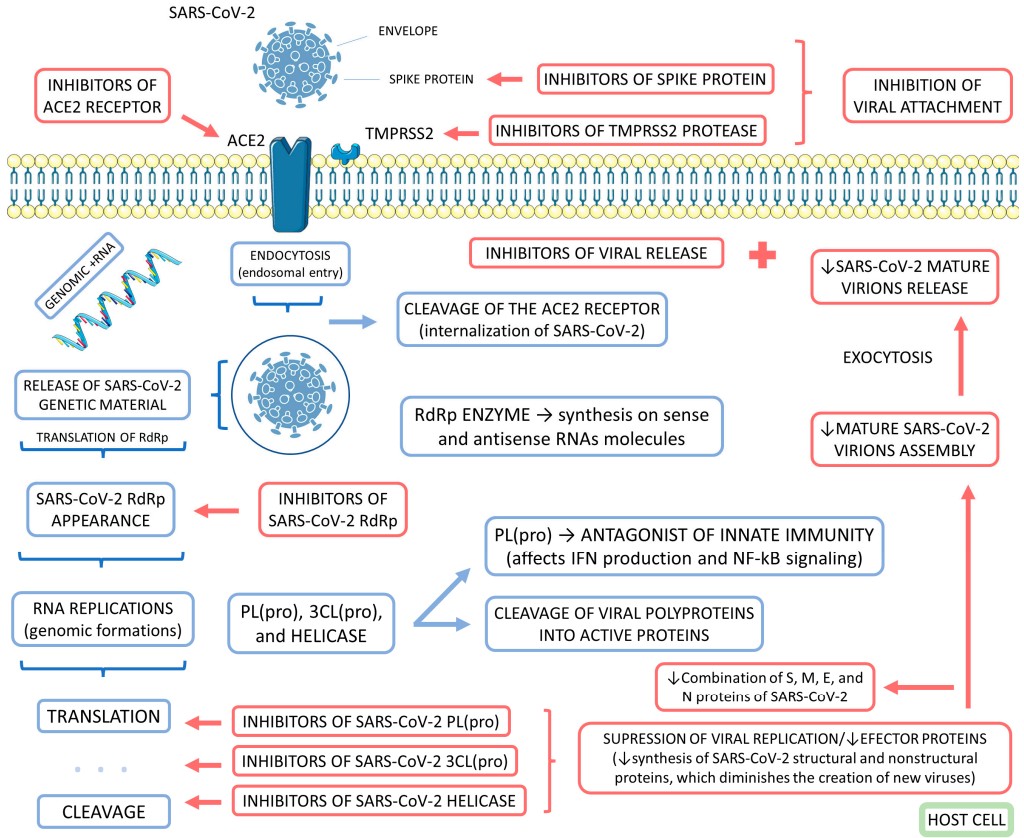

**Figure 3.** Features of SARS-CoV-2 infectious process and potential pathways to inhibit COVID-19 infection. ACE2: angiotensin-converting enzyme 2 receptor, TMPRSS2: type 2 transmembrane serine protease, RdRp: SARS-CoV-2 RNA-dependent RNA polymerase, PL(pro): papain-like proteinase, 3CL(pro): chymotrypsin-like protease, S: SARS-CoV-2 spike protein, M: SARS-CoV-2 membrane protein, N: SARS-CoV-2 nucleocapsid protein, E: SARS-CoV-2 envelope protein, ↓: decrease.

### 3.4. The Role of Medicinal Plants on Influenza

As previously mentioned, influenza viruses are responsible for causing human respiratory infections. Moreover, it is known that infection augments the susceptibility of the infected individual to pneumonia. Therefore, an influenza infection can associate with acute respiratory distress syndrome. Nowadays, several synthetic drugs are used to treat influenza. The existing or in-development drugs Oseltamivir, Zanamivir, Peramivir, and Laninamivir are recommended, due to their inhibitory activity on influenza viruses' neuraminidase. Oseltamivir and Zanamivir are antivirals that are extensively available and have been for decades. Amantadine and Rimantadine are considered influenza A inhibitor drugs. Interacting with viral replication is an effective way to block influenza infection. It is known that influenza only replicates in the interior of the host cells. Therefore, antiviral medications need to penetrate cells without causing cytotoxicity. Furthermore, there is a variable amount of antiviral agents against influenza. The effectiveness of these drugs is limited, principally due to their adverse effects and the presence of antiviral resistance. These limitations contribute to the need for natural antiviral bioactive compounds to treat influenza infections. Medicinal plants and herbs play a fundamental role in treating influenza in some countries around the globe. In most countries, these natural compounds are the leading choice to treat this disease [104–110].

Medicinal plants and herbs target specific actions or features of the influenza viruses to treat the infectious process. Anti-influenza bioactive compounds deactivate or restrain the viruses directly or inhibit influenza indirectly. Some of these actions correspond to regulations of the host immune system, which can amplify the function of the host's immunological properties, leading the human organism to defeat the infection. These actions include inducing interferon production, boosting the activities of the immune cells, stimulating phagocytosis, enhancing macrophage activation, and stimulating the production of IL-1 [102,106].

Bioactive compounds can exert direct anti-influenza activity, protecting against the severity of the infectious process and the possible complications of influenza viruses. The actions of these substances comprise many features, such as influences of viral adherence, viral penetration in the host cells, and effects of viral duplication through the maturation process of viral propagation. Influenza must overcome several protective features of the human organism against infections, such as the innate immune system, other immune responses, and mucociliary clearance of the respiratory tract. Influenza viruses are attracted by the respiratory tract cells, mainly columnar epithelial cells. The entry of the influenza viruses into the host cells depends on the interaction between the viral hemagglutinin and the galactose-bound sialic acid present on the host cells' surface. Viral hemagglutinin is present in the influenza virus's receptor-binding site, and this region attaches to the host cells. After the viral binding, the influenza virions enter the cells via endocytosis by protease cleavage of the viral hemagglutinin, and the virus's replication begins. New RNA genomes are produced, and newly synthesized influenza proteins are created from other RNAs. This process ends with the formation of new influenza viruses from the combination of the viral RNA genome and viral proteins, which leads to the departure of the new virus from the membrane-bond of the cells. Polyphenols are identified as inhibiting proteins and RNA production by the viruses, and present antioxidant and free-radical scavenging properties that can be widely important in the fight against a viral infectious process. These compounds can also be associated with restrained viral adsorption and inhibiting the productive replication processes of influenza A viruses by interfering in the influenza viral budding. Flavonoids can enhance inhibitory actions against the neuraminidase enzyme of influenza viruses. These types of molecules can also inhibit membrane fusion in the viral replication process. Table 3 shows many diverse bioactive compounds with anti-influenza activities [97,104,106,110–114].

### 3.5. Medicinal Plants, COVID-19, and Influenza

Many studies have shown the effects of medicinal plants and isolated phytocompounds against virus infection. In vitro and in vivo studies have evaluated the effects of medicinal plants against SARS-CoV-2 and influenza infections. These studies support further investigations into the antiviral properties of medicinal plants.

Chen et al. [115] conducted an in vitro study to focus on the antiviral properties and other antiviral mechanisms of bioactive compounds derived from *Canarium album* against influenza infections. The cells used in this study were Madin-Darby canine kidney cells. The findings indicated that isocorilagin, extracted from the plant, effectively suppressed the neuraminidase enzymes of influenza A virus strains. Isocorilagin demonstrated safety, affordability, and potential for being utilized as a future anti-influenza drug.

Wolkerstorfer et al. [116] used cultures of human endothelial lung cells, human lung fibroblasts, and Mardin-Darby canine kidney cells to study the effects of glycyrrhizin against influenza A viruses. This study concluded that glycyrrhizin could effectively inhibit influenza A virus uptake by the cells, principally due to mediated interaction between cell membranes with reduced endocytotic activity and diminished virus uptake.

Huang et al. [117] experimented with mice and cells to evaluate the effects of aloin from *Aloe vera* against influenza infections. Aloin could effectively inhibit viral neuraminidase, even when the influenza viruses were Oseltamivir-resistant. From these results, the potency of aloin's anti-influenza characteristics could also be observed in host immunity. With the machinery of the influenza viruses suppressed, aloin could widely boost host immunity against influenza. The immune response was summarized in an augmented hemagglutinin-specific response of T cells against the viral infection. Therefore, aloin can be considered in further human clinical analysis against human influenza infections.

Yu et al. [118] used a culture of mouse aorta smooth muscle cells to evaluate the effects of glycyrrhizic acid (ZZY-44) against SARS-CoV-2. This study showed that ZZY-44 could effectively inhibit the SARS-CoV-2 spike protein, stating that ZZY-44 is efficient against SARS-CoV-2 and can be considered anti-coronavirus. Added to that, in broad-spectrum, the ZZY-44 was considered nontoxic. This study combined computer drug designs with further biological verification.

Park et al. [119], in another study, used diarylheptanoids from *Alnus japonica* against SARS-CoV viruses. This study showed diarylheptanoids' inhibitory effects against SARS-CoV PL(pro). The inhibition of SARS-CoV PL(pro) by the diarylheptanoids opens doors to further analysis of the efficacy of these molecules against SARS-CoV-2 PL(pro).

*Artemisia carvifolia* bioactive compounds were also related to anti-SARS-CoV-2 effects. Arteether, artemether, artemisinin, artmisone, artesunate, lumefantrine, and arteannuin B are the bioactive compounds of this plant. Studies with *Artemisia carvifolia* and Vero E6 cell lines against SARS-CoV-2 showed that the use of the plant blocked the SARS-CoV-2 viral infection due to blockage of post-entry levels. The plant also inhibited the viral RNA and other proteins [104].

Adel Mehraban et al. [36] conducted a double-blind, placebo-controlled, randomized clinical trial to evaluate the effectiveness of violet syrup (*Viola odorata* L. aqueous extract) against COVID-19 infection among 108 outpatients. The main results demonstrated that the plant extract could effectively help to control the classical COVID-19 manifestations such as cough, myalgia, headache, and diarrhea. However, the researchers accepted patients with COVID-19 infection diagnosed by clinical manifestations and/or PCR test and/or radiologist's lung computed tomography (CT) scan report, and not only PCR test. Therefore, the patients' virological clearance was not fully elucidated.

Forouzanfar et al. [37], in a placebo-controlled, randomized, single-blind clinical trial, evaluated the roles of pomegranate juice and SUMAC, a composition of tannins, flavonoids, anthocyanins, isoflavones, terpenoids, and diterpenes, against COVID-19 symptomatology in 178 COVID-19 outpatients. The results showed that patients receiving the pomegranate + SUMAC intervention presented significantly fewer symptoms related to COVID-19 than those patients who did not undergo this treatment. However, the authors described

inadequate patient resilience in post-COVID-19 follow-up. As a limitation, it could be said that although SUMAC presents similar classes of bioactive compounds as pomegranate, the effects of only pomegranate against COVID-19 could have been masked by SUMAC.

Hasanpour et al. [38] conducted a placebo-controlled, randomized, double-blind clinical trial to evaluate the effects of Covexir (*Ferula foetida* oleo-gum) against COVID-19 symptomatology. The results showed that Covexir significantly inhibited many COVID-19-related symptoms such as anosmia, ageusia, myalgia, and cough. However, this study presented a small sample size due to a lack of sample calculation.

Borujerdi et al. [39] conducted a placebo-controlled, randomized, triple-blind clinical trial to evaluate the effects of Zufa syrup (*Nepeta bracteata*, *Ziziphus jujube*, *Glycyr-rhizaglabra*, *Ficuscarica*, *Cordia myxa*, *Papaver somniferum*, Fennel, *Adiantumcapillus ven-eris*, *Viola*, Viper's-buglosses, Lavender, and Iris) in reducing COVID-19 symptomatology. The results showed that the intervention was not significantly different from the placebo and did not exert treatment effects. However, the authors did not use the highest concentration possible of the extract, and the drug dose was low to prevent bad COVID-19 outcomes.

Karimi et al. [40] conducted a multicenter, open-labeled, randomized, controlled clinical trial with COVID-19 patients to evaluate the effects of a polyherbal decoction, using one sachet of the following per day respecting the order: *Matricaria chamomilla* L., *Zataria multiflora* Boiss., *Glycyrrhiza glabra* L., *Ziziphus jujuba* Mill., *Ficus carica* L., *Urtica dioica* L., *Althaea officinalis* L., and *Nepeta bracteata* Benth, and herbal capsules (*Rheum palmatum* L. rizhome, *Glycyrrhiza glabra* root, *Punica granatum* L. fruit peel, and *Rheum palmatum*) for capsule one and *Nigella sativa* L. for capsule two against COVID-19 clinical symptomatology. The results showed that patients showed decreased dyspnea, dry cough, headache, muscle pain, vertigo chills, fatigue, anorexia, sputum cough, and runny nose.

In addition to these studies, other authors have investigated the effects of medicinal plants in clinical trials with humans. An Indian study was the first clinical trial performed on ayurvedic treatment for COVID-19. The main result of this study is a reduction in recovery time in response to treatment and a decrease in changes in serum levels of hs-CRP and pro-inflammatory markers, IL-6, and TNF-$\alpha$. However, the study included asymptomatic or mildly symptomatic patients only; therefore, the clinical criteria could not be used to identify the resolution of the disease, although the exclusion and inclusion criteria were put into practice. Furthermore, the results of this trial cannot be generalized to critically ill patients and patients with comorbidities. Finally, the small sample size requires further studies with larger populations to confirm the findings [41].

A 20-week study of clinical experience with an over-the-counter (OTC) multicomponent "core formulation" consisting of zinc and zinc ionophores; vitamins C, D3, and E; and l-lysine investigated a regimen used in a high-risk multiple exposure population. The formulation offers a low cost, antiviral approach, and the study regimen can serve, at least, as a palliative modality and perhaps a valuable tool in fighting the pandemic. However, the study was neither conducted double-blind nor even blind. In addition, despite their repeated exposures to clinically or test-confirmed COVID-19 carriers, the treatment subgroup did not exhibit cases of COVID-19. In addition, only the demographics of patients who completed the study were reported [42].

A study in China demonstrated that long-term dietary supplementation with lacto-wolfberry in the elderly increases the ability to respond to antigenic challenges without excessively affecting the immune system, strengthening the immune defense in this population and provoking a significantly higher serum response specific to influenza after vaccination. However, whether the lacto-wolfberry effect can be attributed to a single wolfberry, wolfberry compound, or the combinatorial impact of mixing active compounds of fruit or a wolfberry/milk mixture is still unclear. In addition, demographic and prognostic data were not reported [43]. This aspect deserves further investigation.

Another study showed that aged garlic extract supplementation could reduce the severity of cold and flu symptoms. This has been linked to changes in NK and gd-T cell function and reduced secretion of inflammatory cytokines. However, the study had

limitations, such as self-reported diseases, without confirmation of the presence of the pathogen. Colds and flu were not distinguished and the perception of the severity of symptoms by each patient was not recorded; instead, the total number of symptoms was reported as well as the number of days that a specific sign occurred. In addition, the perception of the symptom intensity is modified by diet [44].

Maoto granules, a commercial medical dosage form, are made of four plants: ephedra herb, apricot kernels, cinnamon tree bark, and glycyrrhiza root. One study compared its use to Oseltamivir or Zanamivir in seasonal flu. The statistics did not show significant differences in the total symptom score between groups. During the study period, the viral persistence rates and serum levels of cytokines (IL- 6, IL-8, IL-10, and TNF-$\alpha$) showed no differences between the three groups. However, the study had a small sample size. Moreover, Maoto was not assigned to patients with influenza B virus infection, and the efficacy from Maoto to influenza 2009 pdm has not been confirmed. More large-scale clinical trials are needed to investigate the effectiveness of Maoto against other influenza subtypes for high-risk patients, children, and influenza-related pneumonia [45].

Korean red ginseng (KRG) protects against the contraction of acute respiratory disease (ARI) and can decrease ARI's duration and various symptoms. However, studies are needed with potentially contaminated populations, in addition to children, vaccinated subjects, and immunocompromised subjects, to validate its protective effect [46].

One study has shown that nutritional antioxidant interventions hold promise as a safe, low-cost strategy to reduce the risk of influenza among smokers and other at-risk populations. However, the live attenuated influenza virus (LAIV) vaccine induced the infection in volunteers, which is inherently limited as a model for the disease since the vaccine is designed to have only limited replication in the upper airways. This feature makes the model safe but restricts how generalizable or clinically relevant the study findings are for the community and influenza infections [48].

Another study also used the association of LAIV and broccoli sprout homogenates (BSH). It demonstrated that LAIV significantly reduced NKT and T cell populations and decreased CD56 and CD158b on NK cells, while substantially increasing the expression of CD16 and cytotoxic potential. The supplementation of BSH increased even more induced by LAIV granzyme B production in NK cells than placebo. In the BSH group, granzyme B levels appear negatively associated with influenza RNA levels in nasal lavage fluid cells. This trial showed the same limitations as the abovementioned study [47].

Chinese medicine has shown a complete theoretical system for treating exogenous fevers and illnesses. A Chinese medicine study showed that it has the advantage of treating virus infection, and the effect was accurate, compensating for the shortage of antiviral drugs. In addition, the combination of traditional Chinese and Western medicine is fully viable for cases of influenza A (H1N1), avoiding the resistance phenomenon caused by the mutation of virus genes that antivirals isolated drugs can cause. However, no study blinding was reported, and adequate randomization and more detailed outcome data referring to statistical analyses is needed [49].

A study for the outpatient treatment of influenza with elderberry extract in emergency patients aged 5 years or older presented results that contradicted previous studies and demonstrated the need for further studies, since no evidence was found that the elderberry benefits the duration or severity of flu when compared to placebo. However, the study results were based only on patient symptom reports. In addition, only the first 33 of the 87 patients enrolled in the study had the exact onset time of symptoms, except that they were sick <48 h. Moreover, previous elderberry extract treatment for influenza studies involved patients <48 h after symptom onset. Both studies that started treatment in <24 h showed approximately four days of decreases with time until the symptoms resolved. Furthermore, no complete evidence shows that the elderberry treatment started within the first 24 h of treatment, which may have improved the results [50].

A study with posaconazole for the prevention of influenza-associated pulmonary aspergillosis (IAPA) showed a higher-than-expected incidence of IAPA on admission to the

ICU and incidence below expectations in the population-modified intention-to-treat, which precludes any definitive conclusions about posaconazole as a prophylactic. However, a large number of patients who were excluded from the study due to early IAPA infection questioned the effectiveness of an antifungal prophylaxis strategy initiated at admission to the ICU. Furthermore, it was clear that the required number of patients to be recruited would be at least six times greater than that used according to the sample calculation, which became unfeasible [51].

*3.6. Delivery Systems for Medicinal Plants and Their Derivatives against COVID-19 and Influenza*

The prefix ''nano'' refers to a billionth of a meter in length. In other words, nanomaterials are technology that can be manipulated to achieve specific physical or chemical characteristics to improve synthetic but, principally, natural compounds' bioavailability and health effects. Mostly, size changes bring variations in properties, which are beneficial and offer great potential in treating inflammatory and immunomodulated diseases, such as COVID-19, influenza, and other conditions like rheumatoid arthritis, Alzheimer's and Parkinson's diseases, and inflammatory bowel diseases. Nanoparticles are classified as organic and inorganic and can be designed in different shapes and sizes according to the loading of drugs and the physicochemical properties of the main active substances. Moreover, by using nanocarriers such as nanoparticles, alcohol solutes, and nanoliposomes, natural bioactive compounds and medicinal plants can be loaded and smoothly inserted into certain parts of the body to reduce their side effects, which are already minimal [34,120–122]. Nano-antivirals rely on the small size of viruses to enable interactions with nanoparticles, which offers a practical solution for their treatment.

Curcumin (CUR) is a dietary polyphenol that is exceptionally anti-inflammatory, although hydrophobic, and possesses limited human bioavailability. To overcome CUR's low body absorption and distribution, and to utilize its anti-inflammatory effects against COVID-19, Sharma et al. [123] reported in epithelial cells that curcumin (CUR)-encapsulated polysaccharide nanoparticles (CUR−PS-NPs) effectively inhibited SARS-CoV-2 S protein-induced cytokine storms. Treatment with CUR−PS-NPs significantly attenuated ACE2 interaction with the S protein, and this effect was linked to a reduced NF-kB/MAPK signaling, which in turn contributed to a decreased S protein-mediated signaling and phosphorylation of p38 and p42/44 MAPK and p65/NF-kB, as well as of p65/NF-kB expression. The authors highlighted in their findings the potential of nanostructures in controlling hyper-inflammatory states like COVID-19 and preventing lung injury with SARS-CoV-2-promoted cytokine storm. In addition, Pourhajibagher et al. [124] used CUR-poly (lactic-co-glycolic acid) nanoparticles (CurPLGA-NPs) combined with photodynamic therapy to inactivate SARS-CoV-2 once in plasma using plasma samples of COVID-19 positive patients. The results showed that the CurPLGA-NPs successfully inhibited the virus with photodynamic treatment without significant changes in total plasma protein content, plasma antibodies, and VERO cell viability and apoptosis.

AbouAitah et al. [125] conducted an in vitro study to evaluate the effects of an inorganic–organic hybrid nanoformulation composed of zinc oxide nanoparticles (ZnO NPs) functionalized with triptycene organic molecules and impregnated with ellagic acid (ELG) via noncovalent interactions against DNA and RNA viruses. The results elucidated that ELG alone exerted more cytotoxicity against the host cells than in the ZnO NPs. Furthermore, the ELG ZnO NPs could inactivate H1N1 and HCoV-229E (RNA viruses) and HSV-2 and Ad-7 (DNA viruses) more efficiently. Against RNA viruses, the nanoformulations had therapeutic indexes of 77.3 and 75.7, respectively. In addition, against DNA viruses, the ELG ZnO NPs exhibited therapeutic effects of 57.5 and 51.7, respectively. Specifically, against HCoV-229E, the ELG ZnO NPs exerted direct virucidal actions.

Pilaquinga et al. [126] demonstrated in vitro the efficacy of gold (Au) and silver (Ag) NPs from *Solanum mammosum* L. against SARS-CoV-2 surrogate Phi6 and viral model PhiX174. The plant's antiviral activity was demonstrated using its seed, fruit, essential oil, and leaf extracts, and the leaves were the most effective in inhibiting the viral disease. In

addition, the presence of the plant extracts and their main bioactive compound, gallic acid, on the Au and Ag NPs effectively reduced the metals' toxicity against host adenocarcinoma alveolar basal epithelial cells and human foreskin fibroblasts.

Silymarin is a natural flavonolignan bioactive compound with potent antiviral effects against dengue, hepatitis B, influenza, and HIV viruses. Loutfy et al. [127] used chitosan nanoparticles encapsulating silymarin (Sil–CNPs) as an antiviral treatment against SARS-CoV-2 during in silico and in vitro studies. In this study, Sil–CNPs exhibited low cytotoxicity against Vero and Vero E6 cell lines. In addition, Sil–CNPs revealed great binding energies with SARS-CoV-2 S protein and ACE2, which were $-6.6$, and $-8.0$ kcal mol$^{-1}$, respectively. These results demonstrate a high affinity of Sil–CNPs with the SARS-CoV-2 S protein and high inhibition power. Although the encapsulation with chitosan could effectively augment silymarin bioavailability, the main antiviral activity of the NPs might have been via blocking viral host ACE2 receptors, according to the authors.

Saadh & Aldalaen [128] conducted an in vitro study to assess the inhibitory effects of epigallocatechin gallate (EGCG) combined with AgNPs against avian influenza A virus subtype H5N1, which causes flu infection. The NPs contained 50 μM of EGCG and exerted significant antiviral effects, diminishing the log titer infection by up to 5.7 and 5.6 fold. No cytotoxicity was recognized during this experiment. Although EGCG AgNPS were effective with impact on the host cells, when combined with zinc sulfate these NPs exerted the most potent antiviral activity ever, reducing the log titer of the virus by up to 7.6 times. Saadh et al. [129] also evaluated in an in vitro study the roles of EGCG and zinc sulfate combined against avian influenza A virus H9N2. The results showed that, as in the first experiment against H5N1, the 50 μM EGCG was the most significant concentration to reduce the log titer H9N2 infection. Again, the combination of EGCG and zinc sulfate potentialize the treatment, and no cytotoxic effects were observed against host Vero cells. According to the authors of both studies, combining these elements may impede the transmission of influenza viruses, hinder their replication process in nearby cells, and interfere with microbial resistance by rendering microbial adaptation considerably challenging.

## 4. Materials and Methods

### 4.1. Focal Question

This review was constructed to answer the focused question: ''What are the effects of medical plants on COVID-19 and influenza, and what are the potential delivery systems that could be used to potentialize their antiviral effectiveness''?

### 4.2. Language

Only studies in English were selected.

### 4.3. Databases

This study has included trials found in the National Library of Medicine, National Institutes of Health (MEDLINE–PubMed), EMBASE, Google Scholar, and COCHRANE databases. The descriptors used were COVID-19, COVID, influenza, medical plants, medical herbs, medicinal plants, delivery systems, nanocarriers, or nanomedicine. These descriptors helped to identify studies related to the ingestion of plants and their beneficial role in viral diseases.

### 4.4. Study Selection

This review included studies reporting the potential role of medical plants in patients with COVID-19 or influenza. The inclusion criteria for this study were randomized controlled trials (RCTs), prospective, and placebo-controlled trials. Only full texts were included. The exclusion criteria were animal studies, reviews, non-English studies, case reports, retrospective studies, editorials, and poster presentations. Our aim was to directly evaluate the effectiveness of medicinal plants and phytochemicals in humans and to present

a critical review on the need for clinical trials in this context. Therefore, animal studies were not included in our analysis.

*4.5. Data Extraction*

The search period for this study included the period between July 2011 and February 2023. These studies are described in Table 1. Data were extracted by two independent reviewers (L.T.M. and L.F.L.). A third reviewer was used (S.M.B.) to detect whether any separate article did not meet the inclusion criteria.

*4.6. Quality Assessment*

We consulted the Cochrane Handbook [130] to evaluate each RCT's risk of biases (reporting, selection, and detection). Furthermore, other bias risks in the selection of patients, classification of interventions and outcomes, and missing data were evaluated.

**5. Conclusions**

Medicinal plants and their phytochemicals like bilobetin, glycyrrhizin, sciadopitysin, allicin, eucalyptol, and ellagic acid can inhibit viral adsorption and contain viral replication in many fields of the pathophysiology of influenza and COVID-19 infections. Although other randomized and controlled clinical trials are necessary to evaluate the correct doses, forms, and duration of the treatments, we suggest that using plants that exert antiviral actions against these two diseases should be investigated principally as adjuvants to the traditional synthetic therapies. In addition to the importance of medicinal plants as a source of natural antiviral agents, their bioavailability can be low. Therefore, developing delivery systems such as metal nanoparticles with these compounds can be an alternative solution to reaching infected tissues.

**Author Contributions:** Conceptualization, L.F.L., L.T.M., G.M., V.D.R., R.B.S., V.M.C.S.C., R.K.J. and S.M.B.; methodology, L.F.L., L.T.M., G.M., V.D.R., R.B.S., V.M.C.S.C., R.K.J. and S.M.B.; software, L.F.L. and S.M.B.; validation, L.F.L., L.T.M., G.M., V.D.R., R.B.S. and S.M.B.; formal analysis, L.F.L., L.T.M., G.M., V.D.R., R.B.S., V.M.C.S.C., R.K.J. and S.M.B.; investigation, L.F.L., L.T.M., G.M., V.D.R., R.B.S., V.M.C.S.C., R.K.J. and S.M.B.; resources, L.F.L., L.T.M., G.M., V.D.R. and S.M.B.; data curation, L.F.L., L.T.M., G.M., V.D.R., R.B.S., V.M.C.S.C., R.K.J. and S.M.B.; writing—original draft preparation, L.F.L., L.T.M., G.M., V.D.R., R.B.S., V.M.C.S.C., R.K.J. and S.M.B.; writing—review and editing, L.F.L. and S.M.B.; visualization, L.F.L., L.T.M., G.M., V.D.R., R.B.S., V.M.C.S.C., R.K.J. and S.M.B.; supervision, L.F.L. and S.M.B.; project administration, L.F.L. and S.M.B.; funding acquisition, L.F.L. and S.M.B. All authors have read and agreed to the published version of the manuscript.

**Funding:** This research received no external funding.

**Institutional Review Board Statement:** Not applicable.

**Informed Consent Statement:** Not applicable.

**Data Availability Statement:** Not applicable.

**Acknowledgments:** The authors are grateful to Smart Servier (https://smart.servier.com/, accessed on 6 April 2023) for scientific images that were used in this article under an attribution license of public copyright (https://creativecommons.org/licenses/by/3.0/, accessed on 6 April 2023) and under a disclaimer of warranties. None of Smart Servier's images were changed in the writing of this article.

**Conflicts of Interest:** The authors declare no conflict of interest.

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
