# Peer review of "Exploring the Impact of Herbal Therapies on COVID-19 and Influenza: Investigating Novel Delivery Mechanisms for Emerging Interventions"

_biologics, doi:10.3390/biologics3030009_

Round 1
Reviewer 1 Report
[Biologics] Manuscript ID: biologics-2414212
Title: A Review: Exploring the Impact of Herbal Therapies on COVID-19 and Influenza: Investigating Novel Delivery Mechanisms for Emerging Interventions
- This paper reviews herbal therapies for Covid-19 and influenza, but it is simply too long and uncritically analyzed.
- Please provide inclusion/exclusion criteria for the literature search.
- In the manuscript provide your arguments for excluding animal studies.
- Is it necessary to describe Figures 2 and 3? It is general information.
- The authors should revise section “3. Discussion (3.1 COVID-19, and 3.2 Influenza)” providing a critical data assessment.
- The authors should revise section “2. Results” providing a critical assessment of the data. Were positive controls included.? Was there a dose-response effect?
- The lack of clinical trials is a serious deficiency in evidence-based health benefits.
- Please check the molecules, and draw them following the same style, some do not keep the same format.
I, therefore, recommend major revisions aimed at developing a critical review of Herbal Therapies on COVID-19 and Influenza
Author Response
REVIEWER 1
The authors of this manuscript express their sincere thanks to the Assistant Editor and reviewers for critically assessing this work. The authors have acted upon the recommendations of the editors and reviewers, which have significantly enhanced the quality of this manuscript. We outlined a "point-by-point" response to each comment below. We highlighted all modifications incorporated in the manuscript in yellow.
Comment 1:
This paper reviews herbal therapies for Covid-19 and influenza, but it is simply too long and uncritically analyzed.
Response:
Dear reviewer, we sincerely appreciate your devoted time to reviewing our study. We have carefully considered your suggestions and diligently incorporated them into our manuscript, enhancing its quality. Thank you for your valuable inputs.
Comment 2:
Please provide inclusion/exclusion criteria for the literature search.
Response:
We included Lines 700-703.
Comment 3:
In the manuscript provide your arguments for excluding animal studies.
Response:
We included Lines 703-706.
Comment 4:
Is it necessary to describe Figures 2 and 3? It is general information.
Response:
Dear reviewer, we kindly request to keep the Figures’ descriptions as they are. We appreciate your concern.
Comment 5:
The authors should revise section “3. Discussion (3.1 COVID-19, and 3.2 Influenza)” providing a critical data assessment.
Response:
We included Lines 214-225 and 278-296.
Comment 6:
The authors should revise section “2. Results” providing a critical assessment of the data. Were positive controls included.? Was there a dose-response effect?
Response:
Dear reviewer, we would like to address your concern regarding Lines XXX-XXX. We appreciate your attention to this matter. It is true that several of the studies we examined did not include control patients for comparison purposes. We have taken note of this observation and have included this information in the preceding lines. Thank you for bringing this to our attention.
Comment 7:
Please check the molecules, and draw them following the same style, some do not keep the same format.
Response:
Table 3 has been reformulated.
Comment 8:
I, therefore, recommend major revisions aimed at developing a critical review of Herbal Therapies on COVID-19 and Influenza
Response:
Thank you for your kind words. We value your appreciation and understand the importance of your time as well. We are grateful for the opportunity to you review our manuscript and provide your insights.
Reviewer 2 Report
This review was well written with an interesting topic with good quality of work. The abstract and introduction were carried out properly. I would suggest making the groups of the compounds in Table 3 easier to read. Please check the spelling of pant in Line 406. Please check the reference format. If you include some bioactive compounds in the conclusion section, that will be good.
Author Response
REVIEWER 2
The authors of this manuscript express their sincere thanks to the Assistant Editor and reviewers for critically assessing this work. The authors have acted upon the recommendations of the editors and reviewers, which have significantly enhanced the quality of this manuscript. We outlined a "point-by-point" response to each comment below. We highlighted all modifications incorporated in the manuscript in yellow.
Comment 1:
This review was well written with an interesting topic with good quality of work. The abstract and introduction were carried out properly.
Response:
Dear reviewer, we sincerely appreciate your devoted time to reviewing our study. We have carefully considered your suggestions and diligently incorporated them into our manuscript, enhancing its quality. Thank you for your valuable inputs.
Comment 2:
I would suggest making the groups of the compounds in Table 3 easier to read.
Response:
We have standardized the content presented in Table 3. We appreciate your concern.
Comment 3:
Please check the spelling of pant in Line 406.
Response:
We checked the spelling in Lines 445-448.
Comment 4:
Please check the reference format.
Response:
We reviewed the reference format throughout the manuscript.
Comment 5:
If you include some bioactive compounds in the conclusion section, that will be good.
Response:
We included Lines 717-718.
Reviewer 3 Report
In this review authors approached the topic in a very appropriate way with very relevant figures and tables.The only suggestion is to standardize chemical structure of compounds in table 3. I found this work higly interesting and well treated, so I propose its publication as it is.
Author Response
REVIEWER 3
The authors of this manuscript express their sincere thanks to the Assistant Editor and reviewers for critically assessing this work. The authors have acted upon the recommendations of the editors and reviewers, which have significantly enhanced the quality of this manuscript. We outlined a "point-by-point" response to each comment below. We highlighted all modifications incorporated in the manuscript in yellow.
Comment 1:
In this review authors approached the topic in a very appropriate way with very relevant figures and tables.The only suggestion is to standardize chemical structure of compounds in table 3. I found this work higly interesting and well treated, so I propose its publication as it is.
Response:
Dear reviewer, we sincerely appreciate your devoted time to reviewing our study. We have carefully considered your suggestions and diligently incorporated them into our manuscript, enhancing its quality. Thank you for your valuable inputs.
Table 3 has been reformulated.
Round 2
Reviewer 1 Report
I accept that the work is published